



# Using seasonal forecasts to enhance our understanding of extreme wind and precipitation impacts from extratropical cyclones

Jacob W. Maddison[1], Jennifer L. Catto[1], Sandra Hansen[2], Ching Ho Justin Ng[2], and Stefan Siegert[1]

[1]Department of Mathematics and Statistics, University of Exeter, Exeter, UK
[2]Guy Carpenter & Company Limited, London, UK

**Correspondence:** J.W. Maddison (j.maddison2@exeter.ac.uk)

**Abstract.** Considerable effort is spent at insurance and reinsurance companies to estimate the risk posed by extratropical cyclones (ETCs). Among these risks, strong near surface wind speeds and heavy precipitation can be particularly damaging, threatening infrastructure, human life, and billions of pounds in insured losses. Here, we use nearly 700 years' worth of extended wintertime seasonal forecast model output to estimate the impacts of wind and precipitation associated with Euro-
pean ETCs. Insured losses from winds are estimated with a storm severity index (SSI) and risk of flooding estimated from country-aggregated precipitation totals. Using the Met Office's seasonal forecast model, we follow the UNprecedented Simulated Extreme ENsemble (UNSEEN) method, here applied to ETC impacts. After demonstrating that the model represents ETCs with good accuracy, the likelihood of occurrence of unprecedented ETC impacts are quantified for several countries within Europe. The probability that an ETC will have an impact be more extreme than any observed (i.e. an unprecedented
or unseen ETC impact) is generally between 0.5% and 1.6% for wind and between 0.2% and 0.7% for precipitation across the countries considered. The North Atlantic Oscillation (NAO) is shown to be strongly related to European ETC impact from wind: strongly positive and negative NAO values approximately double and halve the likelihood of an unprecedented wind impact, respectively. The state of the NAO is largely unrelated to the likelihood of extreme cyclone-related precipitation. The dataset created allows for the estimation of impacts from high-return-period storms, which is of great interest to insurance
companies that must be prepared for the potential costs incurred.

## 1 Introduction

Widespread occurrences of strong winds, also known as windstorms and commonly associated with extratropical cyclones, are among the most damaging natural hazards in Europe and therefore one of the most costly (Schwierz et al., 2010; Munich Re, 2015). Costs are incurred, into the billions of pounds, when insurance claims are made against damaged (insured) property.
Together with strong winds, extratropical cyclones can bring extreme precipitation (Easterling et al., 2000; Sibley, 2010; Hawcroft et al., 2012; Pfahl and Wernli, 2012), which also has potential to cause a significant amount of damage (Fenn et al., 2016). The wind and precipitation impacts of ETCs are therefore of great interest to insurance and reinsurance companies. In light of this, much research effort has been devoted to quantifying how much monetary loss historical storms have caused, and what may be expected from storms in the future (e.g. Klawa and Ulbrich, 2003; Leckebusch et al., 2007; Schwierz et al., 2010;



Donat et al., 2011; PERILS AG, 2022). To study the statistics of extreme storms, and estimate their impacts, many hundreds, or even thousands, of years worth of data are needed. In addition, the data must contain realistic representations of ETCs. In this paper, the potential impacts from strong winds and heavy precipitation in ETCs are quantified for several countries within Europe.

The need for a dataset that extends over several hundreds of years, and that is of the spatial and temporal resolutions required
to sufficiently represent a particular feature of interest, is ubiquitous in climate science. Observations may be available that cover the extended time period for certain variables (e.g. temperature), but lack the spatial resolution and coverage normally required, whilst reanalysis products offer (relatively) high spatial and temporal resolution and include many variables, but are limited in the length of their record (typically less than 100 years). Model data is thus often the only option for obtaining such a dataset. In relation to ETCs, climate model output may be too low resolution for reliable cyclone-related surface wind speed and
precipitation information and, due to the long integration of climate model simulations, may contain considerable biases (e.g. Donat et al., 2011; Hawcroft et al., 2016; Tian et al., 2019; Priestley and Catto, 2022; Miao et al., 2023). In catastrophe models, climate model data is downscaled, either dynamically or statistically, to account for this and obtain the necessary resolution to study ETC impacts. The UNprecedented Simulated Extremes using ENsembles (UNSEEN) approach (Thompson et al., 2017) is an alternative for creating the desired dataset: output from many simulations from seasonal forecast models (8 of which are
available through the Copernicus climate data store (CDS, 2023)) are combined to produce a large dataset of model output. As the model is run in seasonal forecast mode, i.e. initialised from reanalysis data each season, run at higher-resolution (compared to typical climate model integrations) and for shorter integration periods, the model output is taken to represent an observation-like data set. The key assumption of the approach is that seasonal forecast systems simulate the feature of interest accurately enough to be considered as an alternative to reality. Previous studies have taken this approach to successfully quantify extremes
in, for example, rainfall (Thompson et al., 2017; Kelder et al., 2020; Kent et al., 2022), temperature (Thompson et al., 2019; Kay et al., 2020) and drought (Squire et al., 2021).

ETCs are another potentially suitable candidate for applying the UNSEEN methodology: there exists some skill in the seasonal prediction of both windstorms in seasonal forecast models (Scaife et al., 2014; Befort et al., 2019; Degenhardt et al., 2023), and precipitation (Thompson et al., 2017; Cotterill et al., 2024). Indeed, an event set of European ETCs was derived
from seasonal forecast data from the ensemble prediction system (EPS) of the European Centre for Medium Range Weather forecasts (ECMWF) in Osinski et al. (2016), an example of the UNSEEN methodology before it was so named. The seasonal forecast system was shown to produce storms with characteristics similar to a reanalysis dataset and hence suitable for creating a large sample of realistic windstorms and facilitating the study of very rare events. The dataset produced in Osinski et al. (2016) was made using 15-day integrations of the ECMWF-EPS (Molteni et al., 1996; Palmer et al., 2007), which at the
time was not coupled to an ocean model. As such, the windstorms produced are not independent from observations and are somewhat constrained to the climatology of the period (e.g. the SSTs). Of course, modifications to windstorm properties and their statistics are found due to the 300-fold increase in sample size. In Walz and Leckebusch (2019), the seasonal forecast system from ECMWF (Seas4) was used to create a dataset of 1,500 years worth of windstorms, which was used to estimate the potential losses from windstorms for four countries in Europe. More recently, Lockwood et al. (2022) used output from a higher



resolution global climate model to create a 1300-year event set of winter windstorms for Europe, again finding good agreement with reanalysis in the number of storms per season and the total storm losses within a season. Several previous studies have also used climate models to estimate extreme wind impacts from ETCs for individual countries (Klawa and Ulbrich, 2003; Karremann et al., 2014), and the European region as a whole (Priestley et al., 2018). Here, we add to this growing body of research by tracking extratropical cyclones in 672 extended winter seasons worth of Met Office seasonal forecast data and

estimating their impacts for 14 countries in Europe. Our approach differs from these prior studies by applying the UNSEEN methodology to ETC impacts for the first time, and by estimating ETC impacts from *both* wind and precipitation.

Estimating the wind impact from ETCs has been the focus of much research. Due to the confidential nature of losses (i.e. how much money an insurance or reinsurance company paid out as a result of the damage from a particular ETC), publicly-available monetary loss data is scarce. In lieu of this, loss indices (often termed storm severity indices (SSIs)) have been introduced and

used to estimate losses based on the winds, typically the maximum wind gust, associated with a given windstorm (e.g. Klawa and Ulbrich, 2003; Leckebusch et al., 2008). SSIs have been shown to correlate well with actual loss data for certain regions within Europe (Klawa and Ulbrich, 2003), or Europe as a whole (Little et al., 2023), and are used routinely by insurers and reinsurers as a proxy for insured losses. The aggregated losses caused by ETCs over a winter season, when cyclones are typically stronger and damages are greater (Munich Re, 2017; Kron et al., 2019), is of particular interest to the insurance

sector. Features of the large-scale atmospheric circulation can impact the aggregated losses in a season. Increased clustering of extratropical cyclones, i.e. occasions when multiple cyclones pass over a region in quick succession, results in higher seasonal loss values from medium-to-high return period storms (Priestley et al., 2018). Fluctuations in the North Atlantic Oscillation (NAO) or East Atlantic pattern (EA), also impact seasonal losses due to windstorms (Walz and Leckebusch, 2019; Lockwood et al., 2022; Priestley et al., 2023), as well as the seasonal prediction of the windstorms themselves (Degenhardt et al., 2023).

The impacts from extreme precipitation in ETCs include water ingress and flooding. ETC-catastrophe models generally include any losses that could be incurred from the storm, typically those resulting from wind and water ingress (with water ingress modelled implicitly rather than explicitly via any precipitation variable). Flooding is usually modelled separately in flood catastrophe models. In flood models, both pluvial and fluvial flooding is accounted for, driven by precipitation not just associated to ETCs but also thunderstorms, for example. Unlike for wind and wind-damage, the relationship between ETC-

precipitation and water ingress and flooding is complicated. Fluvial (river) flooding depends not only on the precipitation totals associated with a given ETC, but also the antecedent conditions in a river basin (e.g. Merz and Blöschl, 2003; Bennett et al., 2018), which can have a strong influence on the likelihood of flooding (Verhoest et al., 2010; Pathiraja et al., 2012; Schröter et al., 2015), whilst pluvial (flash/surface water) flooding, is affected by land use, elevation and drainage, among other things (e.g. Maksimović et al., 2009; Palla et al., 2018; Bulti and Abebe, 2020). For insurers and reinsurers it is nevertheless important

to understand, and have a large sample of, ETC-related precipitation rates for the purposes of developing and evaluating their catastrophe models. We focus here therefore on, and assess the likelihood of, ETC-related precipitation extremes.

There are two main aims of the work presented in this paper.



–   To produce a dataset of extratropical cyclone tracks and their associated wind and precipitation footprints that contains
    several hundreds years worth of data and can therefore be used to study extreme (or high return period) storms and be
useful in the development and evaluation of catastrophe models.

–   To use this dataset to gain insight into the potential impacts of higher return period storms, and in particular to obtain
    an estimate of the probability of unprecedented ETC impacts from both wind and precipitation. Unprecedented here is
    defined as a wind or precipitation impact from an ETC in the seasonal forecast data more extreme than any found in a
    reanalysis.

The article is organised as follows. In section 2, the data underlying the analyses presented are described. The methods that
are followed herein, including the cyclone tracking, SSI calculation, wind-speed to wind-gust adjustment, and precipitation
impact metric, are detailed in section 3. The model fidelity to follow the UNSEEN approach is assessed in section 4. In section
5, the likelihood of unprecedented impacts from wind and precipitation associated to ETCs are quantified, as well as how the
NAO influences these likelihoods. Return periods for ETC-wind impacts are calculated in section 6. The article is concluded
in section 7.

## 2   Data

The primary data used in this study are from a seasonal forecast system that was rerun for 24 years in the past (i.e. seasonal
hindcasts). The seasonal hindcasts used are from the Global Seasonal Forecast System version 6 (GloSea6-GC3.2 system
601, Williams et al., 2018), rerun at the Met Office for years 1993–2016. Fields on a regular Gaussian grid (F128, i.e. a full
Gaussian grid with 128 latitude lines between the pole and equator, which is approximately a $0.7° \times 0.7°$ latitude-longitude grid)
are retrieved for analysis from the Copernicus Climate Data Store (CDS, 2023). Instantaneous values of mean sea level pressure
and horizontal wind components at 10 m are downloaded every 6 hours and used in the tracking algorithm and SSI calculation,
respectively (described in sections 3.1 and 3.2). Daily totals of precipitation are also downloaded for the calculation of the
precipitation impact metric (section 3.3). The seasonal hindcasts initialised in September (on 01, 09, 17 and 25 September of
each year), and run for 215 days, are used herein for each perturbed ensemble member (7 members total). The hindcast runs
therefore cover the extended winter season (October–March). In total, 672 extended winter seasons are analysed (4 hindcast
initiation dates, 24 hindcast years, and 7 ensemble members) with a total of approximately 120,000 storms identified in the
North-Atlantic/European regions (see section 3).

    For verification purposes, the fifth generation atmospheric reanalysis from the European Centre for Medium-range Weather
Forecasts (ERA5, Hersbach et al., 2020) is used. Meridional and zonal wind components at 10 m, as well as the maximum 10-m
wind gust since previous post processing, are downloaded for calculating the SSI. Mean sea level pressure is downloaded for
verification of the storm tracks and daily precipitation totals for verifying precipitation. Data are retrieved for the years covered
by the GloSea6 hindcasts (1993–2016) and at the same spatial and temporal resolutions, namely F128 and 6 hourly. ERA5 is
known to contain some biases in its precipitation (Hassler and Lauer, 2021; Lavers et al., 2022; Bandhauer et al., 2022), though





errors are generally lowest in the midlatitudes during winter (the focus of this study). Precipitation values associated with ETCs also compare well in ERA5 and observations (Lavers et al., 2022), except in mountainous areas. ERA5 precipitation totals are therefore deemed fit for the purpose of this study, namely to serve as an estimate of previously observed ETC-associated precipitation totals.

## 3 Methods

### 3.1 Cyclone tracking

Extratropical cyclone tracks are produced using the objective feature tracking algorithm TRACK of Hodges (1994, 1995, 1999). Mean sea level pressure is used to identify and track cyclones, as in Hoskins and Hodges (2002), with minima in the pressure field (i.e. low pressure systems) identified and tracked at a temporal resolution of 6 hours. Prior to tracking, noise is reduced in the field by truncating it to T63 (triangular truncation with a maximum wavenumber of 63) and planetary-scale features are

removed (those with total wave numbers less than or equal to 5). Extrema in the preprocessed mean sea level field are identified and then joined together to form system tracks. For this work, only tracks that last for longer than two days and travel further than 1000 km are retained, as these are the more mobile systems expected to generate greater impact.

Wind and precipitation footprints are created for each storm track in the seasonal forecast data as follows. For the winds, the wind field is masked in a circle of radius $25°$ centred on each point of a cyclone's track. Then, the maximum wind at each grid

point within the storm footprint across the storm life cycle is retained. This constitutes the storm footprint that is used in the calculation of the SSI. For precipitation, the daily precipitation totals must be assigned to the 6 hourly tracks. The precipitation field of the corresponding day to each track point is masked within $10°$ and the mean precipitation across the track is taken. Any overlapping regions in the masked precipitation along the track points within the footprint on the same day will then be equal to the total precipitation of that day (by taking the mean of the same two daily precipitation totals). The same metrics are

calculated in both ERA5 and GloSea6, enabling a fair comparison between the datasets.

### 3.2 Storm severity index and speed-gust adjustment

The storm severity index (SSI) introduced in Klawa and Ulbrich (2003) is used here, in accordance with the vendor catastrophe models typically used in the insurance and reinsurance sector. The SSI is calculated using the formula

$$SSI = \sqrt[3]{\frac{\sum_{i \in \Omega_0} (v_i - v_0)^3 A_i}{\sum_{j \in \Omega} A_j}}, \qquad (1)$$

where $v_i$ and $v_0$ are the wind field at the $i^{th}$ grid point and the wind threshold that defines extreme winds, respectively. The $\Omega_0$ denotes the set of all grid points within a country, and $\Omega_0$ denotes the subset of these grid points where $v_i > v_0$, and $A_{i/j}$ the area of the $i/j^{th}$ grid point. The cube of the wind speeds is used in the impact metric as this is proportional to the flux of



kinetic energy, and is therefore strongly related to losses (Klawa and Ulbrich, 2003). Thus, the SSI represents both the intensity of the damaging gusts and the geographical extent of the storm.

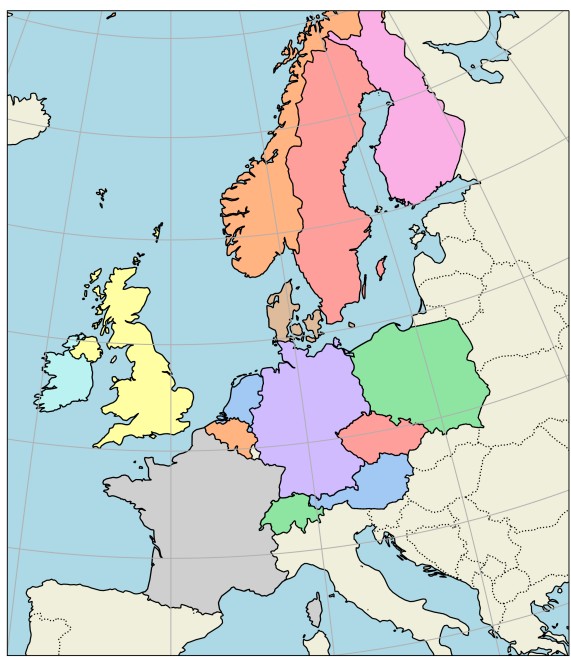

**Figure 1.** The European countries that are included in the analysis presented in this article.

The SSI can be used to aggregate losses over any specified region and time period. Here, the SSI is calculated at the country level for each storm track footprint, as well as aggregated over several countries to provide a European-level SSI value. The countries (and their abbreviations) included here (shown in Figure 1) are Austria (AT), Belgium (BE), Switzerland (CH), Czechia (CZ), Germany (DE), Denmark (DK), France (FR), Finland (FI), Great Britain (GB), Ireland (IE), Norway (NO), Netherlands (NL), Poland (PL) and Sweden (SE), with the European-level SSI an aggregation across these countries. An SSI
value using the cyclone wind footprint as input and following equation 1 is calculated for each individual storm track. The storm footprint SSIs are calculated using the wind gust and wind speed fields in ERA5, and the wind speed fields in GloSea6. The SSIs for each storm footprint in a season are summed to measure the total loss for an extended winter season (termed the annual exceedance probability (AEP) here, following industry conventions). Finally, the wind speed based SSIs are adjusted to values representative of wind gust based SSIs (see next section).

**3.2.1 Conversion from wind speed SSI to wind gust SSI**

SSIs are typically calculated using the maximum surface wind *gust* (e.g. Klawa and Ulbrich, 2003). As these are not available from the GloSea6 archive, a method is developed to calculate SSIs from surface wind *speed* data and convert them to values representative of wind gust SSIs. Recently, Lockwood et al. (2022) similarly estimated wind gust SSIs using the wind speed





output from a climate model. Their approach involved bias correcting the wind speed field to a wind gust field and then
calculating an SSI. The approach used here, described in Appendix A, first calculates an SSI using the wind speed field and
then adjusts to a wind gust SSI. This has the advantage of requiring less data processing, which is useful for such large datasets.

### 3.3 Precipitation impact metric

To estimate the potential hazard represented by extreme ETC-precipitation, we consider country-aggregated precipitation val-
ues for each ETC footprint. That is, for every precipitation footprint created for each ETC track (see section 3.1), the precipita-
tion is summed across every grid point for each of the countries for which the footprint overlaps (i.e. falls within $10°$ of a track
point), and then scaled by the area of that country (as in equation 1). The precipitation metric is therefore country-aggregated,
ETC-associated total precipitation and is calculated in both GloSea6 and ERA5. As the focus here is primarily on *unprece-*
*dented* impacts, there is no need to define a particular threshold or percentile of precipitation as extreme as we compare to the
highest precipitation event in ERA5, which is by definition an extreme event. The precipitation metric is aggregated over the
same countries as the wind metric, i.e. those shown in Figure 1.

### 3.4 North Atlantic Oscillation index

The North Atlantic Oscillation (NAO) index used here is based on the average mean sea level pressure difference between
Iceland (defined here as within the region $63°–70°$N, $16°–25°$W) and the Azores ($36°–40°$N, $20°–28°$W). The average pressure
over the Azores is subtracted from that over Iceland and therefore positive values represent the positive phase of the NAO
(NAO+) and a strong pressure gradient over the North Atlantic. NAO index values are calculated at each (6 hour) time step in
the GloSea6 hindcasts and the corresponding times in ERA5.

### 4 Model fidelity

The GloSea6 seasonal forecast system is first tested in its ability to reproduce key characteristics of ETCs, namely their
associated storm tracks and surface weather impacts, to verify that its output is viable for following the UNSEEN approach.

### 4.1 The North Atlantic storm track and surface weather

Two key features of the North Atlantic storm track are first compared in the GloSea6 hindcasts and ERA5 reanalysis in Figure 2:
the average number of storms passing through each grid point across the extended winter (track density, Fig. 2 left panels), and
the average pressure anomalies of those storms (mean intensity, Fig. 2 middle panels). In GloSea6, the well known features of
the North Atlantic storm track seen in ERA5 are reproduced. Storms are most frequent over the Gulf Stream region (extending
from Newfoundland towards Iceland and northwest Europe), with a gradual reduction of storm frequency to the south and a
sharp reduction to the north associated with Greenland. The number of storms per season is above 60 for much of this region
in both GloSea6 and ERA5. Accordingly, the bias in track density is generally small (Fig. 2(i)) for the majority of the North
Atlantic and European continent. The largest biases are found in the jet exit region, with the pattern in the bias suggesting



the storm track is less tilted in GloSea6 than ERA5 (a bias common in climate model simulations (Pithan et al., 2016)). For

the majority of the countries considered here (Fig. 1), the bias in track density in GloSea6 is small. Considering the bias in track density relative to the value in ERA5 reveals positive relative biases of around 25% extending from Scotland to Denmark (Fig. 2(i)) and negative relative bias of -20% over Scandinavia. The mean intensity of storms is also in good agreement in GloSea6 and ERA5, with the intensity of storms highest in the centre of the storm track region (Figs. 2(b),(f)). There is a marginal overestimation of intensity in northwestern Europe (Fig. 2(j)), but the bias in the mean intensity of storms is generally

small. Relative to the value in ERA5, the relative bias in mean intensity is less than 5% for most of the region considered, except in northwest France and the southeast of Great Britain where the relative bias is positive and approximately 15%.

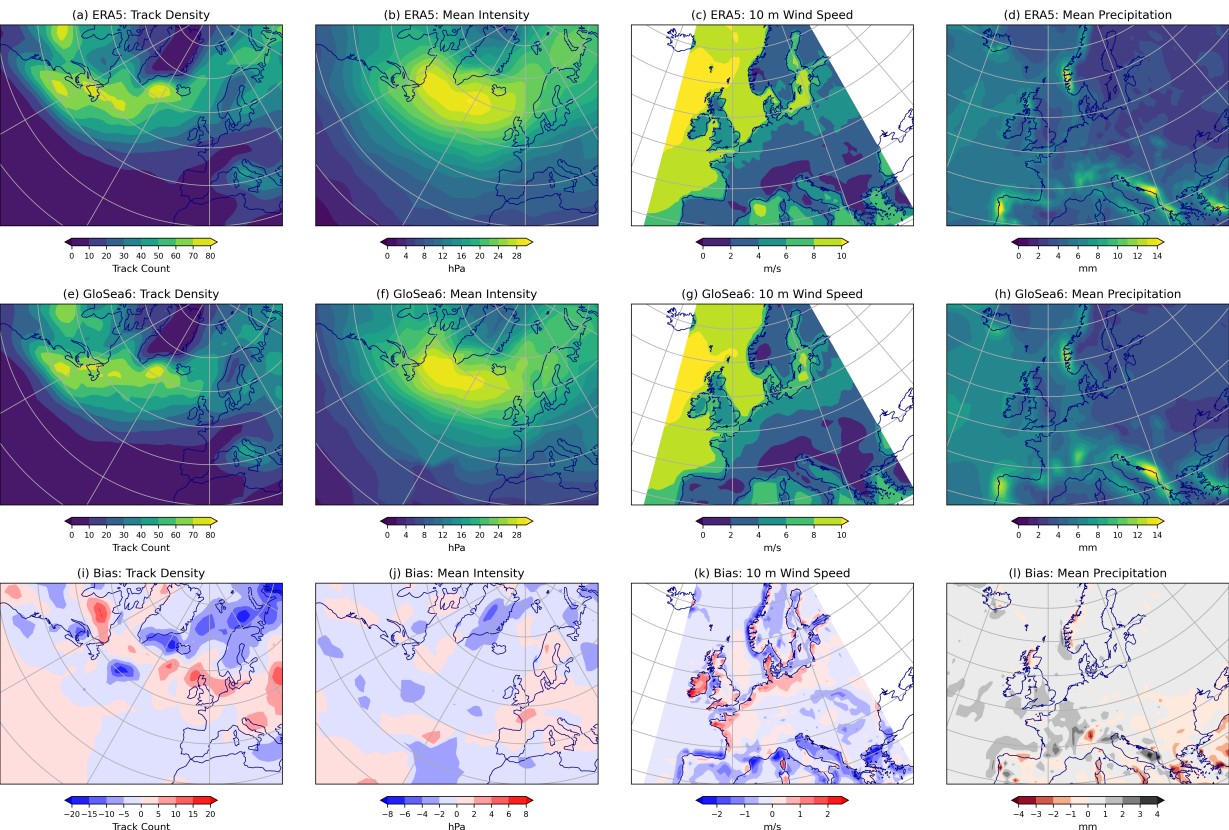

**Figure 2.** A comparison of the GloSea6 seasonal forecast dataset and the ERA5 reanalysis for: the number of storms identified in each grid point during the extended winter (left hand panels), the mean intensity of these storms (where intensity is measured as the magnitude of the central pressure anomaly, second column), the 10 m wind speed (third column), and ETC-associated precipitation (right hand column). Fields are shown for ERA5 (top row), GloSea6 (middle row), and their difference (GloSea6 minus ERA5, bottom row).

The 10 m wind speeds are also compared in GloSea6 and ERA5, as they include the surface impact of the windstorms that reach the European continent and they are the input of the wind impact metric used in this article (section 3.2). The





climatologies of the 10 m wind speed are also qualitatively similar in ERA5 and GloSea6 (Fig. 2(c),(g)). The 10-m wind speeds

are typically larger around coastlines and towards northwest Europe. Smaller scale features identifiable in ERA5 are not present in GloSea6 owing to the lower horizontal resolution, for example around coastal and mountainous regions (Fig. 2(k)). ERA5 retains information from its native high-resolution grid despite having been regridded to the GloSea6 resolution. Nevertheless, it is apparent in Figure 2 that biases in GloSea6 for the storm track and near surface wind speeds are generally small.

Finally, storm-related precipitation is compared in the two datasets. The mean precipitation across all ETC footprints in

ERA5 and GloSea6 is shown in the right hand panels of Figure 2, along with the bias (GloSea6 minus ERA5). ETC-associated precipitation is highest along the west coasts at the end of the storm track (e.g. Norway and Portugal) in both ERA5 and GloSea6 (Figs. 2(d),(h)). ETC precipitation is also typically higher over the oceans than further inland in both datasets. The bias in ETC precipitation is shown in Fig. 2(l). Biases are generally small across Europe. GloSea6 is biased dry for certain western coasts (e.g. Norway and Scotland) and some mountainous areas (e.g. the Alps), and biased wet for Western Europe

(e.g. the UK and France). Wet biases for Western Europe reflect a relative bias of around 15%, which must be noted as a caveat for the results pertaining to the UK and France. The GloSea6 biases for the other countries considered herein are small.

## 4.2  Impact metrics

### 4.2.1  Wind impact

The SSI, and the method to convert from wind speed to wind gust SSI in GloSea6 (section 3.2.1), is now assessed. SSI

distributions for cyclone tracks are shown here for several example countries. Only cyclone tracks for which the SSI value is non-zero, i.e. there is an expected impact in the country, are included in Figure 3. Four different SSI distributions are shown for each country. Two are from ERA5 data: one calculated using the ERA5 wind speed and the other the ERA5 wind gust footprints. The other two are from the GloSea6 data: one showing the direct calculation of the SSI using the GloSea6 wind speed footprints and the other the estimated GloSea6 wind gust SSIs obtained by quantile-mapping.

In general, the wind speed SSI distributions are similar in ERA5 and GloSea6 for the countries included (comparing blue and grey lines in Figure 3), with the mean losses in GloSea agreeing well with ERA5 for the majority of countries. A similar agreement between losses in the ECMWF seasonal forecast system and ERA5 were found in Walz and Leckebusch (2019). The similarity between two distributions can be tested using a two-sided Kolmogorov-Smirnov (KS) test. Doing so reveals that the wind speed SSI distributions are indistinguishable for Belgium and France at a 95% confidence level, and Ireland as well

if testing for difference at a 99% confidence level. It is also evident in Figure 3 that the adjustment successfully maps GloSea6 wind speed SSIs to wind gust SSI values with similar distributions to the wind gust SSIs calculated in ERA5 (comparing purple and black lines in Figure 3). For certain countries there is a greater difference between the GloSea6 and ERA5 wind speed SSIs (i.e. before adjustment), such as Denmark (Fig. 3(b)), or between the adjusted GloSea6 SSI and the ERA5 wind gust SSIs, such as France (Fig. 3(c)). The former is likely due to GloSea6 biases in the wind speed or the number of impactful storms for a

country (Denmark is a region of positive bias in storm track density in GloSea6 (Fig.2(g)). The latter is a result of the quantile





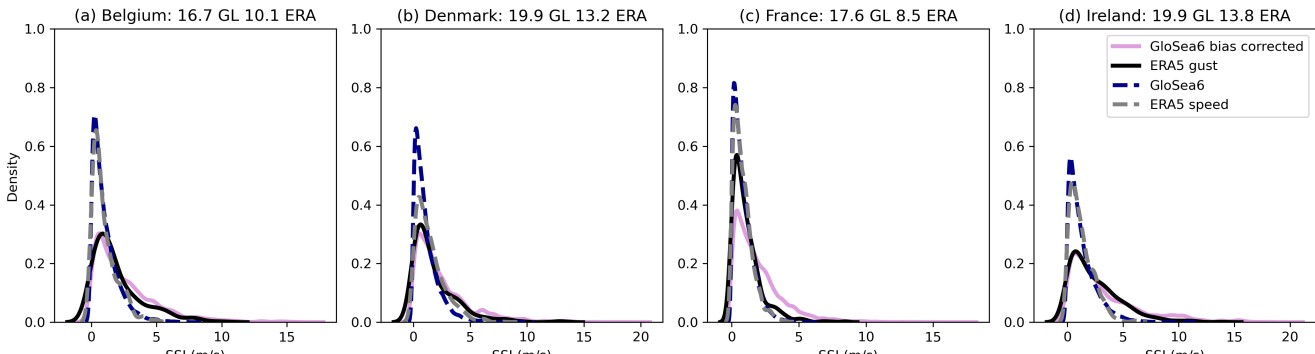

**Figure 3.** Distributions of the storm severity index (SSI) calculated in GloSea6 and ERA5. Wind speed SSIs are shown for ERA5 and GloSea6, together with adjusted wind speed SSIs from GloSea6 (i.e. GloSea6 wind gust SSIs) and wind gust SSIs from ERA5. The SSI distributions are shown for Belgium (a), Denmark (b), France (c) and Ireland (d) as a representative sample of the countries included in this article. Maximum values of the wind gust SSIs are shown in the panel titles for Glosea6 (GL) and ERA5 (ERA).

mapping approach, which is built using the climatology of ERA5 SSIs in a country (not just those assigned to a specific storm track), and therefore may not be representative of the wind storm track defined SSIs presented in Figure 3.

### 4.2.2 Precipitation impact

Distributions for cyclone-related precipitation are shown in Figure 4 for four different example countries. For each ETC pre-
cipitation footprint, the precipitation is summed across the grid points within a country impacted by the ETC and the values for each ETC track included in Figure 4, for both GloSea6 and ERA5. The shape of the distributions for ETC-associated precipitation are similar in GloSea6 and ERA5. This suggests there is no clear bias in GloSea6 for the precipitation metric used and therefore GloSea6 is suitable for studying extremes in ETC-precipitation impacts.

To summarise this section, the output from GloSea6 appears appropriate for the aims of this study, i.e. using the UNSEEN
methodology. Additional tests were performed to ensure that the GloSea6 hindcasts were suitable even towards the end of their evolution (when the model may have drifted towards its own climatology and further from the real atmosphere). Separating the forecasts into their first and second halves and reproducing Figures 2, 3 and 4 does not result in larger departures between GloSea6 and ERA5 (though the storm track biases show somewhat different patterns). We therefore conclude that the entire forecast run is suitable for analysis (not shown), which enables the inclusion of a greater number of storm tracks in the following
analyses and therefore the better estimation of extreme wind storm impacts.

## 5 Unprecedented extratropical cyclone impacts

The UNSEEN approach is designed to inform on events that could plausibly occur at anytime in the current climate, but have not been observed because of the natural internal variability of the climate system and the relatively short observational record.



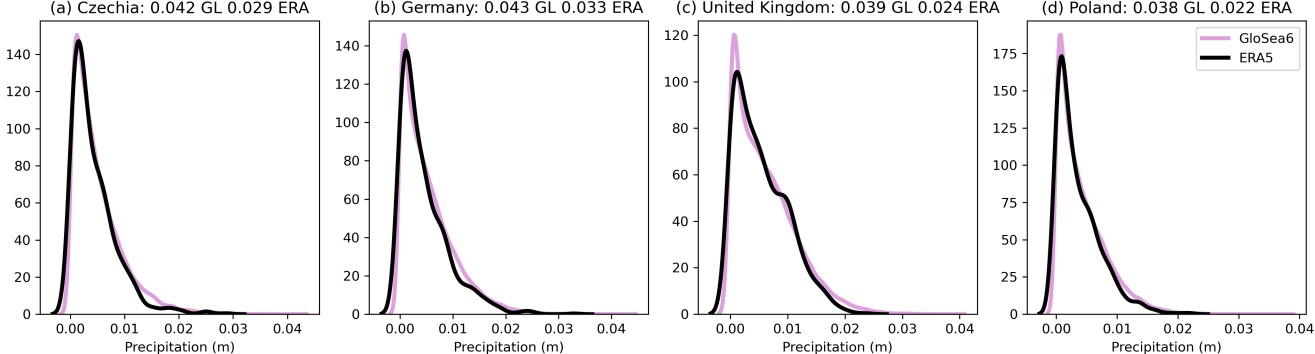

**Figure 4.** Distributions of storm aggregated precipitation in GloSea6 and ERA5. Precipitation distributions are shown for Czechia (a), Germany (b), United Kingdom (c) and Poland (d) as a different sample of countries than that included in Figure 3. Maximum values of the storm-aggregated precipitation are shown in the panel titles for Glosea6 (GL) and ERA5 (ERA).

Here, we investigate whether these so-called unprecedented events, in this case unprecedented extratropical cyclones impacts, are found in the GloSea6 dataset for each of the countries considered and for both wind and precipitation hazards.

### 5.1 Unprecedented wind impacts

The estimated wind-impacts of all of the tracked windstorms (the wind gust SSI values) are shown for GloSea6 and ERA5 for each country of interest in Figure 5 (note that this is the same data as displayed in Figure 3 but for all countries and presented in a way that allows for better visualisation of the extremes). The bulk of the distributions, i.e. the medians and inter-quartile ranges, of the wind gust SSIs are similar for ERA5 and GloSea6 for the countries considered, which suggests that the adjustment used is successfully converting GloSea6 wind speed SSIs to values representative of wind gust SSIs and are hence useful for the evaluation and development of catastrophe models. Unprecedented windstorm impacts are found in the GloSea6 hindcasts for nearly all of the countries. The extent to which the GloSea6 storm impacts exceed the maximum in ERA5 varies by country. For the majority, the most impactful storm in GloSea6 has an impact approximately 1.5 times stronger than the most impactful storm in ERA5 (e.g. Belgium Fig. 5). These storms would result in considerable increases in insured losses.

The number of unprecedented windstorms, and hence the likelihood that one should occur, also depends on the country. The likelihood of an unprecedented storm is calculated by counting all the unprecedented storms for a country and comparing to the total number of storms that have any impact for that country in the GloSea6 hindcasts. The count of storms with unprecedented wind impacts are shown in Figure 7(a), together with the percentage of total storms that this count represents. Counts range from zero for Austria and Finland to more than 600 for Great Britain over the 672 year event set (green circles in Fig. 7). More than 100 storms that would be expected to produce impacts more severe than any in history are simulated in GloSea6 for the majority of countries. Unprecedented windstorm impacts are found, in general, for between 0.5 and 1.2% of all GloSea6





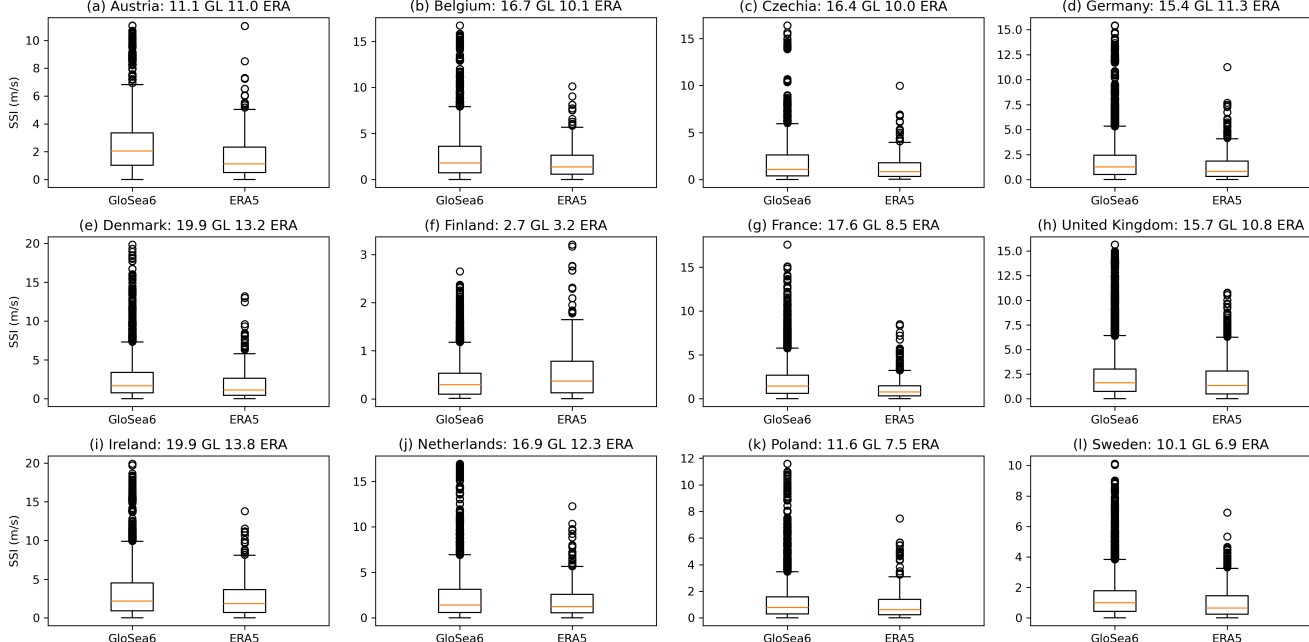

**Figure 5.** Distributions of the wind gust SSI for each country included in the analysis for GloSea6 and ERA5. The values shown in the panel titles are the maximum SSI value in GloSea6 (GL) and ERA5 (ERA).

storms impacting countries within Europe (grey squares in Fig. 5). In other words, there is between a 0.5 and 1.2% chance that a storm that impacts a country will have unprecedented wind impact. Generally, unprecedented windstorms are most frequent for countries in the jet exit region (e.g. Great Britain, Ireland and France). This may reflect the slight GloSea6 biases in this region (Fig. 2). We have less confidence in the results for mountainous countries where the wind speed - wind gust relationship is more variable and the adjustment less effective, such as Austria (not shown).

## 5.2 Unprecedented precipitation impacts

The likelihood of unprecedented precipitation impacts are now quantified. Distributions of the precipitation-impact metric (country-aggregated ETC-associated precipitation) are presented in Figure 6 for each country. As for the wind-impact metric (Fig. 5), the medians (orange lines in Fig. 6) agree well in GloSea6 and ERA5, showing that there are no major biases in ETC-related precipitation in the seasonal forecast model. It is also evident that GloSea6 contains several events in which the expected precipitation impact is more extreme than any observed in the ERA5 historical period. The maximum precipitation impact from an ETC found in the GloSea6 dataset is around 1.5 times larger than the maximum in ERA5. An increase in precipitation from a cyclone of that order would pose a significant flood risk.

The number of events in GloSea6 for which the precipitation is more impactful than any event in ERA5 again depends on the country (Fig. 7(b)). Around 50 unprecedented ETCs are found for Belgium, Czechia, the Netherlands and Poland, whereas



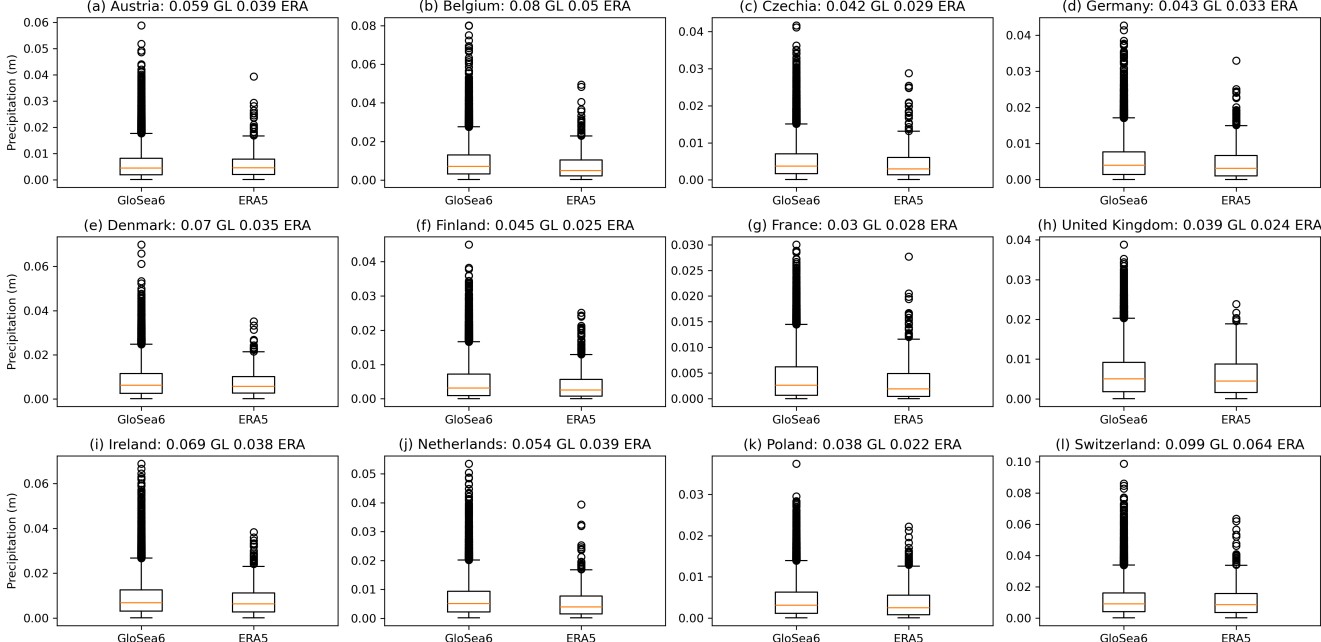

**Figure 6.** Distributions of the precipitation-impact metric for each country included in the analysis for GloSea6 and ERA5. The values shown in the panel titles are the maximum precipitation totals in GloSea6 (GL) and ERA5 (ERA).

there are more than 100 for Finland, the UK, Ireland and Sweden. There are generally fewer unprecedented precipitation impacts than wind impacts across the countries considered. This is reflected in the likelihood of occurrence of an unprecedented precipitation event associated with an ETC, with probabilities typically lying between 0.1% and 0.7%. In words, for an ETC that impacts a country in the current climate there is between a 0.1 and 0.7% chance that the total precipitation associated with the cyclone will be greater than any previously observed ETC (compared with between a 0.5 and 1.2% chance that the impact from wind will be greater).

There is a growing body of research around the compounding impacts of wind and precipitation in ETCs. Systematic studies (e.g. Hillier et al., 2015; Martius et al., 2016; Owen et al., 2021; Bloomfield et al., 2023) suggest that co-occurring wind and precipitation (or flooding) extremes occur with varying frequency across Europe. Co-occurrence is more common along western coastlines in Europe and less so inland, and can often be associated with an ETC (Martius et al., 2016; Owen et al., 2021). The timescale defining co-occurrence can also influence the degree to which the hazards appear to co-occur (Bloomfield et al., 2023). To explore the co-occurrence of the hazards considered herein, we compare, for each ETC track, the wind and precipitation impact metrics for each country. Namely, country-aggregated, ETC-associated precipitation totals are compared to SSI values for ETC tracks for which there is some wind impact within a country (a non-zero SSI value). We find no strong correlations between the wind-impact and precipitation-impact metrics considered (not shown). The impact metrics are both aggregated to country level here. A large SSI value will then be reached on occasions when a large region within a country





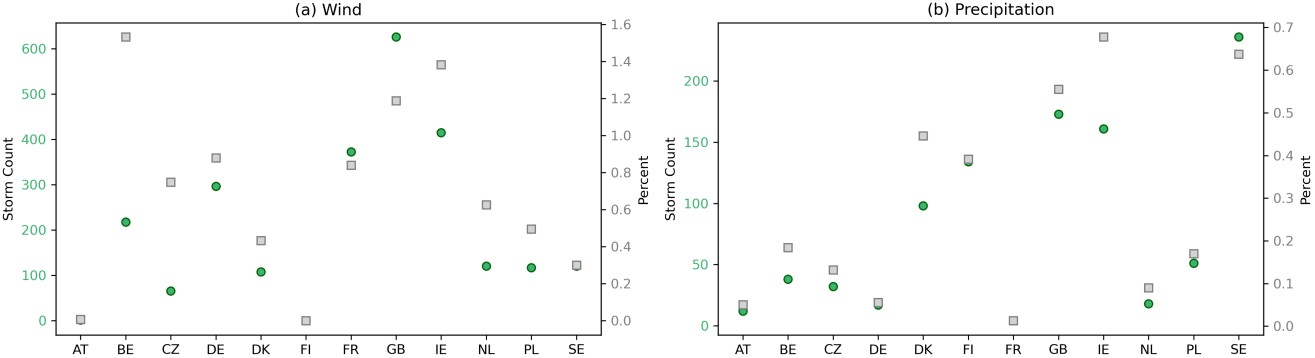

**Figure 7.** (a) The number of ETCs with unprecedented wind impacts in the GloSea6 dataset for each country (green circles, left y-axis). The percent of all of the storms impacting a country that are unprecedented is also shown (grey squares, right y-axis). (b) Same as in (a) but for ETCs with unprecedented precipitation impacts.

experiences high winds, which is more likely for a fast moving ETC. Large precipitation totals for a country may occur when an ETC is slow moving and has more time to precipitate over the same region. This may explain the absence of any relation between the wind and precipitation impact metrics calculated herein. This is similar to the mechanism suggested in Jones et al. (2024), who found that aggregate wind and precipitation extremes are negatively correlated for Western Europe.

## 5.3 Influence of the North Atlantic Oscillation

Previous studies have shown that certain characteristics of the large-scale circulation, such as the NAO, influence the frequency and severity of windstorms, and their impacts, reaching Europe (Donat et al., 2010; Dawkins et al., 2016). Here, we assess if the NOA influences both seasonally aggregated impact totals and the likelihood of occurrence of unprecedented ETC impacts. There is potential for such large-scale circulation patterns to act as predictors of ETC impacts, as they may be better predicted at seasonal timescales than the ETCs themselves (Scaife et al., 2014; Dunstone et al., 2016; O'Reilly et al., 2017). Here, we

consider two ways the NAO relates to storm impact. First, we quantify how the winter mean NAO value relates to total extended winter storm SSI and precipitation totals in both GloSea6 and ERA5. Then, the probability that an unprecedented ETC-impact occurs in the GloSea6 hindcasts is compared during opposite phases of the NAO.

The correlation between the total wind impact in a season (i.e. the AEP) and the winter mean NAO value is shown in Figure 8(a) for GloSea6 and ERA5. In ERA5, the AEP is correlated with the mean NAO value: when the NAO index is

more positive the total losses are increased (Fig. 8a). Correlations are slightly weaker for countries further east, e.g. Poland and Sweden, where the influence of the NAO is typically weaker. The correlation is also weaker in France. France sits at the midpoint of the dipole of typical NAO impacts, and the anomalies in relevant variables remain relatively consistent for oppositely signed NAO values and the correlation with the AEP is lower. There also exist correlations between the AEP and NAO index in the GloSea6 hindcasts. The correlations are weaker in GloSea6 compared to ERA5, though they are statistically





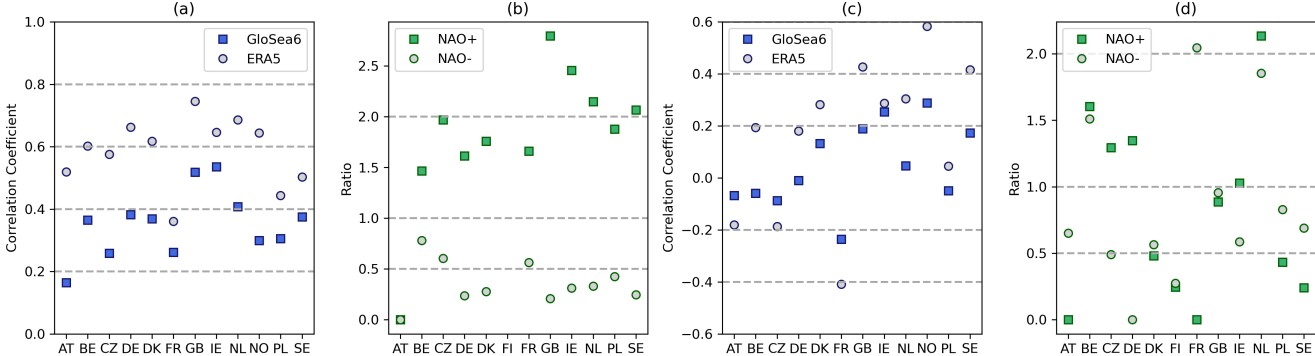

**Figure 8.** (a) Correlations between the annual exceedance probability (AEP) and the seasonal mean NAO values for each country considered in the GloSea6 and ERA5 datasets. (b) The change in likelihood in GloSea6 of occurrence of unprecedented ETC wind impact given a strongly positive (NAO+) or strongly negative (NAO-) NAO index, defined as above the 90th percentile of NAO values or below the 10th, respectively. Dashed lines in (b) represent no changes in likelihood (1.0), a doubling in likelihood (2.0) and halving in likelihood (0.5). (c) As in (a) but correlating the seasonal mean NAO with seasonal totals of ETC-precipitation. (d) As in (b) but for the ETC precipitation impact metric.

significant. The relationship between expected seasonal losses and the winter mean NAO found here in GloSea6 are similar to those previously found in high resolution climate model simulations (Dawkins et al., 2016; Lockwood et al., 2022), and other seasonal forecast models (Walz and Leckebusch, 2019). Seasonal mean NAO values are less related to season precipitation totals (Fig. 8(c)). In ERA5, most countries, particularly those towards Western Europe, exhibit a weak to moderate positive correlation. The seasonal mean NAO value is negatively correlated to seasonal precipitation for Austria, Czechia and France.

GloSea6 exhibits a similar pattern but weaker magnitudes of correlations across the countries as ERA5. The seasonal mean NAO value reflects the jet stream's strength in a season, with high NAO values corresponding to a stronger jet stream and faster-moving ETCs. These systems will typically have higher wind speeds but may not bring large precipitation totals to countries as they have less time to precipitate, explaining the low correlations between the seasonal mean NAO and seasonal ETC precipitation totals. High NAO seasons will however be associated with an increased number of storms which may explain the

moderate correlations for certain countries, particularly those in northwestern Europe, e.g. Ireland, Norway and Great Britain.

     The change in likelihood of an unprecedented ETC impact occurring in GloSea6, given the state of the NAO, is shown in Figure 8(b) for wind and Figure 8(d) for precipitation. The probability of an unprecedented wind impact approximately doubles on occasions when the NAO index is anomalously high (above its ninetieth percentile), though this is country dependent and the change in probability varies from 1.5 times more likely to nearly 3 times. Probabilities are stronger for countries further

west, such as Great Britain, where the probability of an impactful storm (i.e. a storm with a non-zero SSI in a country) being unprecedented goes from 1 in 83 to around 1 in 30, and Ireland, where the probability changes from 1 in 71 to 1 in 29. Conversely, on occasions when the NAO index is strongly negative (below its tenth percentile) the likelihood of an unprecedented wind impact drops to roughly half its typical value for most of the countries. A similar change in the *mean*




impact of windstorms given the state of the NAO was shown in Walz and Leckebusch (2019). Knowledge of the mean state

of the NAO in a hindcast run could therefore be useful in preparedness for not only the expected impacts from strong ETCs
(as shown previously (e.g. Walz et al., 2018; Lockwood et al., 2022)) but the potential for an impact more extreme than ever
observed. If this can be well forecast at longer lead times, and better than the prediction of individual ETCs, the skill horizon of
aggregated extreme windstorm impacts could be extended. The state of the NAO has less of a relationship with unprecedented
precipitation impacts. The number of ETCs with unprecedented precipitation impacts is generally smaller than those with

unprecedented wind impacts. Subsetting these based on highly positive or negative NAO years results in a very small sample
and therefore unreliable statistics on which no robust conclusions can be drawn.

## 6    Return periods for wind impacts

In this section, we focus on the wind impact of ETCs. Insurers are required, under the Solvency II directive[1] to cover losses with
99.5% confidence. This corresponds to being prepared for a 1-in-200-year event. It is therefore of great interest to insurance

companies to have accurate estimates of the potential impact of a 200-year return period storm. One of the aims of this paper
is therefore to provide another estimate of the impacts from the most extreme wind storms and in particular a 200-year return
period wind impact. With this in mind, we calculate storm return periods in the GloSea6 dataset and compare the GloSea6
results to those obtained from ERA5. The dataset extends the return period of storm impact that we can estimate to 672 years.
(This is compared to around 80 years for the full ERA5 dataset, or the 50 years (1972-2022) used in Figure 9.)

365        The wind gust SSI values, as a function of return period, are shown for ERA5 and GloSea6 in Figure 9 for Austria, France,
Great Britain and the Netherlands. The historical storms in ERA5 are extrapolated using a generalised Pareto distribution (GDP
fit), which is then used to estimate the SSI for higher return periods. This serves as a comparison to the SSI values for high
return periods obtained here using GloSea6. The shape of the SSI-return-period curve is of most interest here, in particular for
the right tail of the distribution that identifies the extreme storms, rather than the specific SSI values represented in GloSea6

(which will contain some model bias). As such, an additional correction is performed whereby the GloSea6 return period curves
are shifted in the vertical so that the 10-year return period value matches that in ERA5. The shape of the curves estimated by
GloSea6, as above, are therefore retained, and by assuming the shape in GloSea6 is accurate, an estimation of the potential
impacts of the most extreme storms is possible.

        For the majority of countries, the SSI values for different return periods generally agree well in ERA5 and GloSea6. This is

not the case for Austria (Fig. 9(a)), where the SSI-return-period curve in GloSea6 lies above that in ERA5 for the lower return
period storms (despite being bias corrected to match ERA5 for the 10-year return period level). There is a large discrepancy
between GloSea6 and ERA5 for Austria because it is a mountainous country, and the relationship between wind speed and
wind gust is less well corrected for by the quantile mapping approach. For mountainous regions, peak wind gusts can reach
very high values but sustained wind speeds are typically much lower, and so a high wind gust value may not necessarily

---

[1]The European Union Solvency II directive dictates losses must be covered with 99.5% confidence (https://eur-lex.europa.eu/legal-content/EN/ALL/?uri=CELEX:32009L0138 [last access February 2024]



**Figure 9.** SSI curve against return period (note the x-axis is on a log scale), for the GloSea6 event set (orange) and ERA5 event set (black) for (a) Austria, (b) France, (c) Great Britain and (d) Netherlands. SSI values for ERA5 are here calculated using the hourly 10 m wind gust since previous post-processing at 0.25 x 0.25 degree resolution as follows. Event dates in the ERA5 event catalogue are extracted for storms between 1972-2022 and the SSI is calculated based on the 3-second peak gust using a 20 m/s threshold using equation (1). A Generalized Pareto Distribution (GPD) is fitted to the reference catalogue following the Peaks Over Threshold (POT) method. The median fit (black solid lines) and its corresponding uncertainty (black dashed lines) are shown. The uncertainty of the GPD fit results from the selection of the threshold and parameters of the fit. This is reported to 95% confidence intervals. The thresholds for the POT method are selected based on the 85-95th percentiles of SSI values. For GloSea6, SSI values for the full event set are shown in orange crosses, while the orange lines show 20 samples of these, each of 50 randomly selected seasons. An additional bias correction is included here whereby the GloSea6 curve is shifted vertically so that the SSI value at a return period of 10 years is equal to that in ERA5.



respond to a high wind speed value (as is typically the case for other regions), which renders the quantile mapping technique
used here less effective. As the majority of countries considered here are not mountainous they do not suffer from a similar
issue. This is evidenced by the more closely matching SSI-return-period curves for France, Great Britain and the Netherlands,
where for the bulk of the distribution the shape of SSI return period curves agree well in ERA5 and GloSea6. In France
(Fig. 9(b)), the GloSea6 SSI values match those in ERA5 for even the most extreme storms, adding confidence to the estimates

of windstorm impacts currently being used in the insurance industry. For higher return period storms in Great Britain (Fig. 9(c))
and the Netherlands (Fig. 9(d)), GloSea6 SSIs are lower than the extrapolation of ERA5 and outside of the GPD fit uncertainty
estimate. This is due to an apparent flattening of the SSI curve with increased return period. The quantile-mapping is converting
the GloSea6 SSI values to a distribution with a flattening tail as the tail also begins to flatten in the ERA5 wind gust SSI
distribution (the GPD fit also flattens off, though not as quickly). A similar flattenning of the return period curve for SSIs was

found in climate model data in Lockwood et al. (2022), with SSIs increasing quicker for return periods less than around 500
years, after which the SSI increase is more gradual. Return periods for storms in Great Britain and Germany did not exhibit a
similar flattening at higher return periods in the event set created in Walz and Leckebusch (2019).

One of the strongest storms in history in Great Britain is storm Daria (with an SSI value around 13.0 it is one of the rightmost
black marks in Fig. 9(c)), which impacted the UK in January 1990 and caused more than £7 billion of insured losses (as if 2023

values). The flatness of the GloSea6 SSI curve suggests that the most extreme storms in that dataset have similar SSI values and
that there is considerable uncertainty in their return period. For a severe storm such as Daria, datasets that cover longer periods
than current historical records are needed to more accurately estimate the return period. This kind of information, comparing
the most impactful windstorms from recent memory and better constraining their return period, can be used in the insurance
sector for comparison with catastrophe model output, and ultimately better estimations of the total insured losses that would

need to be paid out in the event of a very high return period storm. This can be done for each country individually, and for
many of the other well known historical storms, to reduce the uncertainty on the impacts that could be expected from a storm
that may occur at anytime in the future.

## 7   Conclusions

Output from the Met Office's seasonal forecast system (GloSea6) was used here to create nearly 700 years of extended win-

tertime extratropical cyclone (ETC) tracks. Unprecedented impacts from these cyclones, caused by their associated wind and
precipitation footprints, are quantified using the UNSEEN methodology (Thompson et al., 2017; Osinski et al., 2016). Apply-
ing the UNSEEN methodology to ETC impacts allows the first quantification of the likelihoods of more extreme impacts than
ever observed to be experienced for several European countries. The probability that an ETC impacting a country will have
an unprecedented wind impact is generally between 0.5% and 1.6%, for the northern/central European countries considered

herein. An unprecedented precipitation impact from an ETC is less likely, with the probability between 0.2% and 0.7%. The
GloSea seasonal forecast system is known to represent windstorms with reasonable accuracy (Degenhardt et al., 2023), and
additional analyses presented here confirmed its fidelity in windstorm representation. A better understanding of extreme ETC





impact is of key importance to society, as they are among the most damaging and the most costly natural hazard impacting Europe (Munich Re, 2015; Fenn et al., 2016) and are expected to become more severe in the future climate (Schwierz et al., 2010; Priestley and Catto, 2022).

The GloSea6 seasonal forecast system was shown to represent extratropical cyclones and metrics of their wind and precipitation impacts well enough to follow the UNSEEN approach (Thompson et al., 2017) for estimating extremes. Firstly, the number of storms identified within a season, and their mean intensities were shown to be similar in the two datasets. The distributions of an SSI computed from the tracks' wind footprints, as well as distributions of a precipitation impact metric, based on country-aggregated ETC-associated precipitation totals, were again closely matched in the hindcasts and reanalysis. Considering all of this, the GloSea6 output was deemed suitable for creating around 700 years worth of cyclone tracks and studying their impacts. Similar approaches were followed in Osinski et al. (2016), Walz and Leckebusch (2019) and Lockwood et al. (2022) to create datasets containing a large sample of windstorms associated with ETCs, with this work adding adding to this body of research. The model used here, the method to convert from wind speed to wind gust SSIs and the additional inclusion of precipitation impacts, distinguishes the presented results from these previous studies.

To study ETC wind impact, a quantile-mapping adjustment approach was used to convert values of a wind-speed SSI to those representative of a wind-gust SSI. The latter is most commonly used in the insurance industry, as is the aggregation of windstorm impacts to the country level as is done here, and hence the results presented are directly comparable with catastrophe model output, and immediately available for their development and evaluation. A similar quantile-mapping was used in Lockwood et al. (2022) to create their windstorm dataset. In Lockwood et al. (2022), quantile-mapping was used to bias correct wind speeds to wind gusts on which an SSI was calculated, whereas here quantile-mapping was used to covert wind-speed SSIs to wind gust SSIs. Both methods appear to produce usable wind gust SSIs from model wind speeds. Datasets with only wind speed output available and not wind gusts, such as GloSea6, other seasonal forecast systems, as well as climate model simulations, can therefore be used in comparison with catastrophe model output, giving an opportunity for the creation of even larger datasets of windstorms and their impacts. The impacts of extreme storms, estimated with the adjusted SSI, were quantified and compared to the reanalysis. Hundreds of ETCs with wind impacts greater than any known in the historical period, so called unprecedented impacts, were found for the majority of countries considered, with between 100 and 600 unprecedented windstorms identified for most countries. This reflects a probability of a storm being unprecedented between 0.5% and 1.6%.

The potential impacts from precipitation in ETCs include water ingress and flooding, with the two being modelled by separate catastrophe models used in the insurance industry and flooding also driven by other extreme precipitation drivers, such as thunderstorms. Owing to data availability in GloSea6, we focus on the ETC-precipitation peril and create a metric to quantify the severity of the hazard posed by precipitation within ETCs. Actual flood risk will depend on other factors such as antecedent river levels and annual precipitation totals (e.g. Bennett et al., 2018), in the case of fluvial flooding, and instantaneous rainfall rates and drainage conditions in the case of pluvial flooding (e.g. Bulti and Abebe, 2020). ETC-precipitation information is still valuable for insurers and can be used to estimate the potential losses expected from flooding that may occur due to the extreme precipitation rates they bring. The precipitation impact metric is calculated as the country-aggregated ETC-



associated precipitation totals. This metric agrees well in ERA5 and GloSea6, and unprecedented ETC-precipitation events
are also identified in GloSea6. They are less common than their wind counterparts, with the probability of an ETC having an
unprecedented precipitation hazard between 0.2% and 0.7% in general. No strong correlation is found between the ETCs' wind
and precipitation impacts in this study.

The phase of the North Atlantic Oscillation (NAO) is shown to relate more to wind impacts than precipitation impacts in
ETCs. Seasonal losses from winds are positively correlated with the seasonal mean NAO index, in agreement with previous
studies (e.g. Walz et al., 2018; Lockwood et al., 2022; Priestley et al., 2023), and the likelihood of an unprecedented wind
impact is approximately doubled when the NAO index is strongly positive (and roughly halved when the NAO index is strongly
negative). This offers a potential avenue for extended predictability of winter ETC wind impacts, as the NAO is reasonably
well predicted many months in advance (Scaife et al., 2014; O'Reilly et al., 2017). The NAO had less of a clear relationship
to the precipitation impact metric, with seasonal ETC-precipitation totals only weakly correlated to the seasonal mean NAO
index.

The Climate Data Store archive contains many more thousands of years worth of data from different seasonal forecast
models and their ensemble members. Further work analysing these data would improve the quantification of unprecedented
ETCs and their potential impacts from wind and precipitation presented here, and in several previous studies (e.g. Osinski et al.,
2016; Walz and Leckebusch, 2019; Lockwood et al., 2022), thus improving confidence in the estimates of the most extreme
present-day ETC impacts and hence, for insurance and reinsurance companies, estimate of the amount of capital that must be
held to be able to pay the claims that would arise from such storms. This is crucial knowledge for the insurance sector.

*Code and data availability.* ERA5 and GloSea6 data are freely available from the Copernicus Climate Data Store (CDS, 2023). The cyclone
tracking code is available from the authors upon request.

## Appendix A

The SSI used herein is intended to identify damage inducing winds on approximately 2% of days, following Klawa and Ulbrich
(2003). The 20 m/s wind gust threshold value used in the wind-gust SSI in this and other studies was defined based on the
2% of days criteria for wind gust values and insurance company practises in Germany (Klawa and Ulbrich, 2003), but has
been widely adopted as a suitable value to use for other European countries and is the threshold commonly used in catastrophe
models run by insurance companies. To calculate an SSI from wind speeds it is therefore required to define a wind speed
threshold for the SSI that is equivalent to the 20 m/s wind gust threshold. To do this, the percentile that 20 m/s represents
in the ERA5 wind gust climatology is identified, and then the wind speed threshold ($v_0$) is defined as the wind speed value
at that percentile in the climatological wind speed distribution. A 20 m/s wind gust is typically between the $96^{th}$ and $98^{th}$
percentile for the countries considered, with a mean of the $97.5^{th}$, (supporting the use of a 20 m/s wind gust threshold across
countries as done in catastrophe models). The wind speed thresholds used in the wind speed SSI are calculated separately for



each country considered to account for potentially different wind gust-wind speed relationships across countries, as well as
in ERA5 and GloSea6 separately to account for potential GloSea6 biases in near surface wind speed. The threshold values
used for calculating the wind speed SSI in ERA5 and GloSea6 are listed in Table A1, together with the standard error in the
threshold value estimate.

| Country | AT | BE | CH | DE | DK | FI | FR |
|---|---|---|---|---|---|---|---|
| ERA5 | 6.1 ±0.4 | 9.0 ±0.5 | 8.5 ±0.5 | 8.4 ±0.4 | 9.5 ±0.6 | 10.0 ±1.1 | 8.5 ±0.4 |
| GloSea6 | 5.9 ±0.4 | 9.8 ±0.5 | 8.8 ±0.5 | 9.7 ±0.5 | 11.7 ±0.6 | 9.8 ±0.4 | 9.5 ±0.4 |
| Country | GB | IE | NL | NO | PL | SE | |
| ERA5 | 8.9 ±0.5 | 8.4 ±0.5 | 9.4 ±0.7 | 6.4 ±0.5 | 9.4 ±0.7 | 8.2 ±0.5 | |
| GloSea6 | 10.3 ±0.4 | 10.8 ±0.4 | 11.0 ±0.5 | 8.4 ±0.6 | 10.1 ±0.6 | 8.9 ±0.4 | |

**Table A1.** Threshold values ($v_0$, m/s) used in the calculation of the wind speed SSI together with the standard error of the estimate of the percentile used in the threshold calculation. Values are shown for ERA5 and GloSea6. Standard errors are calculated as the standard deviation of wind speed threshold values calculated across evenly sized samples (1000 data points) of the data.

The final step in the procedure for converting SSI values derived from wind speeds to values representative of wind gust SSIs is quantile-mapping. This correction is required for the data produced to be used in the evaluation and development of
catastrophe models, and for a better comparison with the previous literature. A quantile-mapping approach (Thrasher et al., 2012) is used here to map the wind speed SSIs to wind gust SSIs. Quantile maps are made for each country in Europe using the ERA5 wind gust SSI and ERA5 wind speed SSI. The approach works as follows. First, cumulative distribution functions (CDFs) of the model (ERA5 wind speed SSI) and reference (ERA5 wind gust SSI) data are estimated over a set of regularly spaced quantile intervals. Then, to convert a wind speed SSI value $x$, the percentile of $x$ is calculated in the model CDF
estimate. The value in the reference CDF at this percentile is found and taken as the adjusted wind speed SSI, i.e. the wind gust SSI estimate. The ERA5 quantile maps are used for both the ERA5 wind speed SSIs and GloSea6 wind speed SSIs to enable the study of extreme SSI values using GloSea6. A quantile map by definition maps the extremes of the model data to the reference data and thus the adjusted values cannot exceed the maximum value in the reference data on which they are made (i.e. the maximum ERA5 wind gust SSI). By using the ERA5 based quantile maps for GloSea6, we retain the possibility
that adjusted SSI values in GloSea6 can exceed the maximum in ERA5 (when the GloSea6 wind speed exceeds the maximum ERA5 wind speed).

*Author contributions.* JWM wrote the manuscript and performed the majority of the analyses. CHJN performed the return period and storm clustering analyses. All authors contributed to the design of the study as well as providing feedback and comments on previous drafts of the manuscript.



*Competing interests.* The authors declare they have no competing interests.

*Acknowledgements.* The authors wish to thanks Kevin Hodges for his support setting up and running the cyclone tracking algorithm. This work used JASMIN, the UK's collaborative data analysis environment (https://jasmin.ac.uk, Lawrence et al., 2013). Guy Carpenter & Company Limited funded this research.



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
