# Peer review of "Using seasonal forecasts to enhance our understanding of extreme wind and precipitation impacts from extratropical cyclones"

_EGUsphere, 2025_

## Author Comment (AC1)

We thank the reviewers for their time and effort reviewing our manuscript. In the following, the reviewers' comments appear in black and our responses in blue.

**Reviewer 1**

Review of Using seasonal forecasts to enhance our understanding of extreme wind and precipitation impacts from extratropical cyclones, by Maddison et al.

This paper proposes to use an ensemble forecast dataset as a surrogate to investigate the impacts wind and precipitation from extra-tropical storms. The paper uses a UK MetOffice product (GloSea6) and compares it the the ERA5 reanalysis. The paper focuses on the statistical analyses of impact indices related to precipitation and wind speed. The authors investigate the relation with the North Atlantic Oscillation (NAO).

The paper is well organized and the idea of using ensemble forecast data as surrogates of reanalyses is very appealing.

Major comments

Why not use the ensemble members of ERA5 (rather than the mean)?

We use ERA5 as an estimate of reality and a point of comparison for the seasonal forecast output. The ensemble members in ERA5 could indeed be used to estimate its uncertainty, but it is known that the maximum uncertainty values for midlatitude storms are low (less than 1 hPa) (https://confluence.ecmwf.int/display/CKB/ERA5%3A+uncertainty+estimation#:~:text=The%20uncertainty%20estimates%20for%20ERA5,horizontal%20and%201h%20temporal%20resolution). We would therefore expect the spread among the ERA5 ensemble to be small and no additional understanding obtained from using the ensemble and so choose to only include the ERA5 reanalysis.

The authors determine empirically the probability of exceeding the record (highest value) of ERA5 in the GloSea6 ensemble, after having verified that the two datasets yield similar probability distributions (Figures 3-6). In principle (B. Arnold et al., Records, Wiley, New York, 1998, Ch. 2), if the record in ERA5 is obtained in, say, N=75 years, then the probability to exceed this record (say in GloSea6) is 1/(N+1). This is close to the empirical estimates that the authors find. Therefore, an important finding of the paper (increasing the data size increases the probability of exceeding the record) is actually fairly trivial from the statistical point of view (i.e. one can get it from a paper-pencil computation).

We will add a comment detailing that the increases we find are inline with what you would expect statistically from increasing the sample size. We would however expect a similar increase across the countries from a statistical standpoint but we observe quite considerable differences between them. For countries that we find have a larger probability of unprecedented storms, the storms observed in the historical record may not be as severe as they potentially could have been (and a more extreme storm happening in the near future is therefore more likely). For countries that have a lower probability, an extreme storm has already been observed and an unprecedented storm is less likely. We will add discussion relating to this in the revised manuscript.

What is not trivial, but undiscussed, is the strange behavior of SSI data in Finland, for which the GloSea6 distribution is much lower than the ERA5 distribution, although the core distributions look similar. Any idea?

In line with our response to the previous comment this suggests that Finland has already experienced several very extreme storms in recent decades, and an unprecedented storm occurring in the current climate is unlikely.

The return level plots in Figure 9 are probably wrong (the curves should start from the same return period). What are the Pareto distribution parameters? Computing return level plots from a Pareto

distribution fit is potentially tricky, especially because the SSI values are conditional to the occurrence of storms, and not on a time axis. This is where conditioning on an NAO index (for example) could be more useful. A clarification on how the GPD fits are obtained is necessary.

The GPD fit was performed to the SSI values assuming independence with time, using the peaks-over-threshold method. The thresholds for POT method are selected based on the 85-95th percentiles of SSI values, and then the uncertainty of the GPD fit results from the selection of the threshold and parameters of the fit (and reported to 95% confidence intervals).

The curves do not start from the same return period simply because the GPD fit curves are plotted at the discrete RP (2, 5, 10, etc.) for which the SSI is nonzero.

This information will be added to the methods section of a revised manuscript.

Minor comments
What the authors call "loss" is actually the value of wind or precipitation indices. This does not pertain to actual insurance losses, and might be misleading.

This is true, but the indices (particularly the SSI) are widely used in the industry as proxies for loss and correlate reasonably well which actual loss values, which is why we inherited this nomenclature. We will highlight this in the revised manuscript.

Figures 3-4 seem redundant with figures 5-6 (same information?).

Figure 3 shows that the bias correction method is working suitably, which is not shown in Figure 5. Figure 4 can be removed in a revised manuscript without losing any information.

L. 181: The NAO index on sub-daily increments might not be super relevant (it is generally used on monthly time scales) because of the spatial variance of the low and high pressure systems.

We do use the sub daily NAO index values but primarily to calculate winter mean NAO values, as the reviewer is correct in that the daily index would be very noisy and not relevant for European weather. We will make this clear in the methods section and main text.

Sec. 5.3: The methods section should explain into more details what is done in Figure 8. How exactly are computed the ratios? Are there uncertainties?

We can expand the methods section to include this. For panels (a) and (c) we simply plot the correlation between the seasonal AEP (i.e. the summed losses across the season for wind and precipitation separately) and the seasonal mean NAO index.

For panels (c)/(d) we create two additional samples of cyclones. We extract those that occur during strongly negative or positive NAO values (based on the 10th and 90th percentiles of daily NAO index values) and then calculate the likelihood of finding an unprecedented storm in these two subsamples (by comparing those with SSIs bigger than the maximum in ERA5 with the total number in each subset). The ratio is then calculated by comparing to the likelihood of an unprecedented storm occurring in the full data set.

This information can be added to the methods section in a revised manuscript.

L. 467: The cyclone tracks are produced by the TRACK algorithm of Hodges, and obviously by K. Hodges himself (cf. Acknowledgments). How can the algorithm be distributed from the authors, since K. Hodges does not distribute it? The cyclone tracks should be made available without a request to the authors.

The cyclone tracks produced for this project are proprietary to Guy Carpenter (as the sole funders of the research), which means they cannot be made freely available unfortunately. However, the tracking code can be made available on request to Kevin Hodges (it is not available to download

but the algorithm is made freely available via email to Kevin Hodges). Furthermore, ERA5 and GloSea6 are also publicly available so the work is reproducible.

**Reviewer 2**

Overall the paper describes some good work but I feel some of the emphasis is misplaced. It has a slight feeling of trying to sell something with claims about the significance for the insurance industry. Particularly on the windstorm aspect, I would like to see some clearer discussion on what the results show about the likelihood of 'unseen' storms and how that fits with existing understanding.

More specific comments:

Line 36-37 seems dismissive of GCMs and seems to imply that the unseen dataset would not require downscaling or bias correction, which is not substantiated in the paper.

We wanted to highlight that GCM biases are expected to be bigger than in seasonal forecast models (as seasonal forecasts are typically run at higher resolution and they are initialised from observations each year). We will rewrite this sentence to reflect this.

69-73 SSI discussion is brief and uses outdated references. Their limitations and alternatives could be better considered. Previous works include Pinto 2012 and Karremann 2014, then Moemken 2024 showed that while SSI or similar metrics are able to rank storms, they are not an estimator of loss in extreme cases.

The discussion relating to SSIs can be expanded and updated with more recent research. For our purpose, comparing the likelihood of a storm more severe than any observed in ERA5, a ranking of storms is sufficient. We are mostly interested in the number of storms that are unprecedented rather than the specific loss/SSI value of the storms. We can make this clear in a revised manuscript.

110 - is the dataset resolution .7deg? This is quite low. Population weighted SSI at this resolution may not classify storms very well and will introduce biases towards core Europe.

The data is at this resolution. We do not use a population weighted SSI for this reason and instead just consider the windspeed in the SSI calculation. As we only consider country-wide impacts our definition of the SSI provides a country level indicator of loss potential. Users, as in reinsurers, can use this information to estimate more local scale impacts using their own models/methods.

170 - This seems a key step relegated to an appendix.

This was placed in the appendix as it is not crucial for interpreting the results, and may be only necessary for those more specialised in the subject area or looking to reproduce our results. Some information can be added to the main text that briefly describes the method.

196 - 60 storms per season seems high if we are intending to focus on extremes

60 is the total number of storms in the extended winter season (ONDJFM), which is 10 storms per month or between 2 and 3 storms per week. These are not the extreme storms but all tracked storms so this number seems reasonable.

Fig 2 - could this analysis focus on the extreme events?

The purpose of Figure 2 is to demonstrate that the characteristics of storms/storm tracks are reasonably represented in GloSea and therefore suitable for studying extremes. It would not be straightforward to show these biases for only extremes storms as we focus on country level impacts. An extreme storm in one country will not necessarily be extreme in another. This would make selecting storms to include in an "extreme only" version of figure 2 difficult. We therefore think including all storms in Figure 2 is a fairer, cleaner comparison.

210-213 - The biases are small but coastal features etc are crucial in loss modelling.

We can emphasise this point here in the text. Unfortunately some biases are unavoidable due to the resolution of the seasonal forecast data. As we aggregate our loss indices to the country level the impact of these small scale coastal features will be minimised.

Fig 3 - Again, I would like to see more focus on extremes, but acknowledge this comes later.

The purpose of this figure is to demonstrate that the bias correction is working sufficiently. As we bias correct all storms we prefer to show the entire distribution here.

264/fig 5 - visually I don't find this a very helpful way of showing the extremes particularly eg UK. I'm not critical of the data shown here but this feels like a key result that could be illustrated better.

This figure was designed to highlight the outliers in GloSea6, while at the same time demonstrating the bulk of the distributions are similar in ERA5 and GloSea6. The authors cannot think of a better/clearer way to demonstrate this and therefore opt to leave the figure as is.

279 - reference to figure 5 I believe should be to figure 7?

Thanks for pointing this out, it has been corrected.

310-313 - this feels well-reasoned and good demonstration of an existing hypothesis even if limited to a per-storm time window.

We thank the reviewer for this comment.

Fig 7 - Are some of these very high? 600 storms in 697 years in UK is around 1 per year. Lack of strong storms in AT is attributed to the gust quantile mapping in a mountainous country. What about Switzerland? It is mapped in figure 1 but does not seem to be included in the results. Also for Finland is this not a tracking issue with storms not persisting sufficiently far East?

Switzerland was indeed omitted from the analysis because we found that the bias correction method performed poorly for Switzerland. This was because the shape of the distributions of wind speeds and wind gusts were more different for Switzerland, likely because of the complex topography making the relationship between windspeed and wind gust more non-linear. This is also the case for Austria but the bias was less severe so we retained Austria in the analysis.

There is a negative bias in track count for Finland which may be indicative of storms not persisting far enough east as the reviewer suggested. The 600 storms identified as unprecedented for the UK may also be partly impacted by the biases in GloSea6 (which is biased slightly positively for track count and mean storm intensity).

We will mention these points in a revised document.

Section 5.3 OK with all this. Good discussion.

We thank the reviewer for this comment.

371 & 374 - I understand what is being done but it reads to me that you state the 10 year RP is matched then claim that the agreement across most RPs is good which then seems obvious.

Thanks for pointing this out, we were referring to the shape of the return period curves here. This will be corrected.

Fig 9 - particularly for UK this now seems to present a conclusion that is contradictory to figure 7. we now see that at eg 100 years RP all the GloSea6 storms are below the extrapolated ERA5

lines, whereas previously it was claimed that the dataset had a very large number of storms above the highest historic observation. I presume this is due to the 10yr RP scaling but this feels contradictory to the claims in the introduction that the dataset would not need re-calibrating due to biases.

The storms in GloSea6 are below the extrapolated ERA5 data, but there are many more extreme than the highest in the observed ERA record (all those to the right of the final black dot denoting ERA5). Therefore the figures are not contradictory but reveal that our dataset suggests that the more extreme storms we will see are not expected to be as severe as those estimated from a statistical extrapolation of ERA5. This point will be made clearer in the text.

396 - Can we, for example. estimate an RP for Daria from the dataset? This is an example of how the discussion could focus on how this dataset enhances or adds to existing understanding of extreme windstorm risk. Existing works include the XWS and C3S catalogues. Flynn 2024 (admittedly in pre-print) has some good review of these. Alongside the other works referenced around lines 57-64.

This type of analysis can indeed be done with the dataset created. For example, comparing the storm with the highest SSI for the UK in ERA5 (the black dot furtherest right in Fig.9 c), which has a return period of the length of the ERA5 data used, with the return period curve predicted by the GloSea dataset, we can see that storms with a similar SSI are found for return periods up to around 200 years. Therefore a Daria like storm is unlikely to reoccur in the near future. This type of analysis can be included in a revised manuscript.

Thanks for pointing to the Flynn 2024 reference, we will discuss the relevance for our work in the revised manuscript.

Appendix A: This seems quite brief, could it not just be in the text? I would like to see some more recent thinking on SSI included beyond the 2003 paper.

See response to previous comment regarding the decision to make this an appendix. The discussion relating to SSIs can be extended in the introduction/methods section.